# Seven Decades of Spontaneous Forest Regeneration after Large-Scale Clear-Cutting in Białowieża Forest do not Ensure the Complete Recovery of Collembolan Assemblages

**Marek Sławski *** and **Małgorzata Sławska**

Faculty of Forestry, Warsaw University of Life Science-SGGW, ul. Nowoursynowska 159, 02-776 Warsaw, Poland; malgorzata_slawska@sggw.pl
* Correspondence: marek_slawski@sggw.pl

**Abstract:** The long-term effects of large-scale disturbance on forest ecosystem processes and structure are poorly understood. To assess the effects of large-scale clear-cutting on the taxonomic and functional structure of collembolan assemblages, 18 plots were established in the Polish part of Białowieża Forest. All plots, situated in a mixed *Tilio-Carpinetum* broad-leaved forest, had eutrophic Cambisol developed on rich glacial deposits. The Collembola assemblages in the stands that had naturally regenerated on large-scale clear-cuts performed at the beginning of the 20th century were compared to those in old-growth forests (i.e., the endpoint of stand development following stand-replacing disturbance). Collembolans, one of the most numerous soil microarthropods, are successfully used to assess the consequences of forest management and ecosystem restoration. Our study tested whether seven decades of spontaneous forest development after large-scale anthropogenic disturbance ensures the complete recovery of the soil Collembola. Using complementary taxonomic and life-form approaches, we provide evidence that the collembolan assemblages associated with the tree stands that had spontaneously developed in large harvesting plots distinctly differed from those in old-growth deciduous forests in this region despite seven decades of regenerative forest succession. The species diversity of the assemblages in the naturally regenerated tree stands was significantly lower, and their life-form structure was noticeably different from those in the reference forests. Moreover, the shift in the functional group structure of the collembolan assemblages in the stands that had regenerated after clear-cutting indicates that their activity seven decades after disturbance is concentrated mainly on the decomposition of the litter in the upper layers, whereas the processes controlled by these organisms in the deeper soil layers are not fully restored.

**Keywords:** old-growth mixed deciduous forest; harvesting; natural regeneration; soil microarthropods; taxonomic and functional structure

## 1. Introduction

In recent decades, European forests have been faced with large-scale disturbances caused by windstorms and pests [1,2]. In view of the predictions of an increasing frequency of natural disturbances resulting from global climate changes [3,4], the ability to escape from such events in the future is unlikely. Since they cannot be forecasted and their results cannot be diminished, forest management and forest planning must involve the development of new, more flexible approaches to adequately respond to these increasingly difficult challenges. Setting aside a part of destroyed forest stands for natural regeneration is one of the discussed concepts [5]. With respect to protected areas, such an approach is common; with respect to managed forests, this strategy is the subject of controversy,

because its ecological and economic consequences are unknown. Therefore, studies to elucidate the dynamic pattern exhibited by naturally regenerating forests are urgently needed, especially because the objects for such studies are currently available.

Białowieża Forest, inscribed by UNESCO United Nations Educational, Scientific and Cultural Organization on the World Heritage List, includes a complex of lowland forests that are characteristic of the Central European mixed forests terrestrial ecoregion. The area has exceptional conservation significance due to the scale of its old-growth forests, which include extensive undisturbed areas where natural processes occur. However, the wasteful exploitation of these precious forest stands has taken place over various periods in history. Such activity in Białowieża Forest occurred mainly during World War I in the years 1915–1918 by German occupants and shortly after that, in 1924–1929, by The Century European Timber Corporation. During these periods, 4 million and 2.5 million cubic meters of wood were cut down, respectively, mainly via clear-cutting [6,7]. As a result of The Century European Timber Corporation activity, extensive clearings 100 m in width and 1 km long with individual trees of low wood quality, largely spruce, and hornbeam were left aside. Only a small proportion of them were later reforested by the planting of Scotch pine, while in the majority of the plots, pioneer species such as birch (*Betula pendula* L.), aspen (*Populus tremula* L.), and willow (*Salix caprea* L.) have spontaneously established. At present, in the Polish part of Białowieża Forest, naturally regenerated tree stands in the large-scale clear-cutting area occupy over 6 thousand ha, mainly in the Browsk and Hajnówka forest districts (2.36 thousand ha and 2.24 thousand ha, respectively) [8].

The ecological consequences of clear-cutting have been well studied, because such information is critical to the development of sustainable management practices. In general, harvesting whole trees or more intense practices, such as stump extraction and site preparation involving forest floor removal, decreases the deadwood volume, herb vegetation, and moss and organic soil cover. It has been shown that all these changes have negative effects on the diversity and abundance of soil microarthropods [9–15]. Unfortunately, the majority of these studies involved short-term experiments. A few existing long-term studies have shown that the taxonomic structure of mesofaunal communities generally remained modified, with a lower density and species diversity and altered species composition five [16], 10 [17], 17 [18], and even 20 years after clear-cutting [19]. The revealed persistence of community disruption suggests that the recovery of soil microarthropods after harvest is very slow; therefore, long-term investigations are needed to assess ecological rotation, i.e., the recovery of a full suite of ecosystem services (according to Rousseau et al. [15]).

In temperate and boreal forests, springtails (Collembola, Hexapoda) represent some of the most numerous soil microarthropods involved in organic matter decomposition, nutrient cycling, and soil microstructure improvement [20]. A large number of collembolan species occurring in litter and soil contribute essentially to the biodiversity of forest ecosystems, and as a result enrich the diversity and functionality of belowground food webs [21–24].

During long-term succession, collembolans respond to various environmental variables, and therefore can indicate progress in ecosystem development through changes in community structure [25–27]. Several studies have used Collembola as a model group because of the high sensitivity of this taxon to soil disturbances and environmental factors, such as humidity, temperature, pH, and humus form [20]. Moreover, springtails are a very diverse group that occur in all types of soils, inhabit both litter and the deeper soil layers, use a wide range of food sources, and serve as prey for many predators. Four functional guilds of Collembola related to life forms (sensu Gisin [28]) are recognized by Potapov et al. [29]: (1) epigeic plant and microorganism consumers, which affect the dynamics of the first stages of litter decomposition; (2) epigeic animal and microorganism consumers, which regulate the population density of microorganisms and microbivores and possibly affect wood decay rates; (3) hemiedaphic microorganism consumers, which control microbial communities and affect the physical structure and mineralization rates of litter; and (4) euedaphic microorganism consumers, which affect nutrient uptake by roots and regulate the microbial community in the rhizosphere and soil organic matter decomposition. Therefore, detailed information regarding the life-form structure of

Collembola assemblages provides deep insight into the ecological processes in the soil and the ability to better understand the changes in their course in disturbed ecosystems. Functional trait analysis is increasingly being recognized as an important tool in soil ecology and forest management research [30–33]. This approach can be especially useful in the assessment of the recovery of forest ecosystems, both those that have regenerated naturally and those that have established by replanting or afforestation. There have been several studies using Collembola as an indicator of the progress of the later type of succession [27,34–40].

To our knowledge, no long-term studies have considered the species-level response of Collembola to natural forest succession, because the spontaneous development of trees on large clear-cuts is rarely observed in managed temperate forests, where artificial regeneration as a result of planting predominates. The aim of our study was to take advantage of the forest stands that have spontaneously developed after large-scale clear-cutting in a lowland forest area with high biodiversity where natural processes predominate. The focus of this study is the Collembola assemblages that have developed over many decades in forest stands subjected to only minor anthropogenic pressure. We speculate that large-scale clear-cuts involving the removal of biomass have a profound negative impact on Collembola inhabiting the forest litter and soil. We hypothesize that the number of species and the abundance many of them are reduced on clear-cuts, and that the taxonomic and functional community structure of the mesofauna is distinctly altered. We also assume that the recovery of the soil mesofauna in large-scale harvesting plots is very slow, and that for some species with low locomotive ability, it may take many decades to re-establish viable populations in disturbed areas.

Finally, we aim to answer the following questions:

Can seven decades of spontaneous forest development on large-scale clear-cuts lead to the complete recovery of the taxonomic structure of collembolan assemblages?

Are the soil processes driven by diverse collembolan life forms fully restored in naturally developed stands?

## 2. Materials and Methods

### 2.1. Study Area

The Białowieża Primeval Forest (52°43′ N, 23°50′ E; altitude between 134 and 202 m a.s.l.) is located in northeastern Poland on the Bielsk Plain. Glacial ablation moraines composed of sandy, clayey, and loamy soils occupy 75% of the reserve area, with the remaining area composed of organic deposits, primarily peat. The climate of the region is continental, with warm summers and cold winters. The total annual precipitation is 641 mm (85% falling as rain), the mean annual temperature is 6.8 °C, and snow cover occurs for a duration of 92 days.

To assess the differences in collembolan assemblages induced by large-scale clear-cutting, a total of 18 plots were established (Figure 1). All plots had eutrophic Cambisol soil developed on rich glacial deposits and potentially represent mixed broad-leafed *Tilio-Carpinetum* forest (Natura 2000 code 9170). The vast majority of forest stands naturally regenerated after the wasteful exploitation of moderately humid deciduous forest habitat [8]; therefore, all study plots were located in this forest type. Six tree stands that had spontaneously developed on large-scale clear-cuts (three in the Browsk Forest District and three in the Hajnówka Forest District) were chosen for the study. As reference sites, six mature linden-oak-hornbeam forests of the same habitat type were selected, three in each in the above-mentioned forest districts, as well as six old-growth forests in the Orłówka Protection District in Białowieża National Park (formerly Strict Nature Reserve). The latter plots were located in the central part of the reserve, and included the most valuable old-growth stands in Białowieża Primeval Forest [41].

The study plots in the reserve (Res) were located in the following management units: 284Ba, 314Bd, 317Ab, 318Df, 320Cc, and 373Db. The old-growth stands were very diversified, with a complex horizontal structure that usually consisted of three strata of trees (Figure S1). The first tree layer

was composed of mixed stands of oak (*Quercus robur*), Scots pine (*Pinus sylvestris*), Norway spruce (*Picea abies*), hornbeam (*Carpinus betulus*), and maple (*Acer platanoides*) in various proportions with an admixture of linden (*Tilia cordata*), silver birch (*Betula pendula*), and aspen (*Populus tremula*). The coverage of the first tree layer varied from 30% to 70%, most often approximately 50%. The ge of the oldest trees varied from 120 to 250 years old (approximately 190 on average). The second tree layer consisted of hornbeam (10%–80%), maple (0–60%), and spruce (0–60%) with an admixture of linden and oak. The total coverage of the second layer of trees varied from 10% to 80%, and was most often above 50%. The understory consisted of young hornbeams (10%–80%) and linden (0–20%) with an admixture of spruce and maple. The total coverage of the understory layer varied from 10% to 100% (most often approximately 50%). The ground flora was typical of *Tilio-Carpinetum* forests and was dominated by *Anemone nemorosa*, with usually more than 20% coverage, accompanied by *Lamium galeobdolon*, *Galium odoratum*, *Stellaria holostea*, and other species. All tree stands in the Strict Nature Reserve have been under strict protection since 1921.

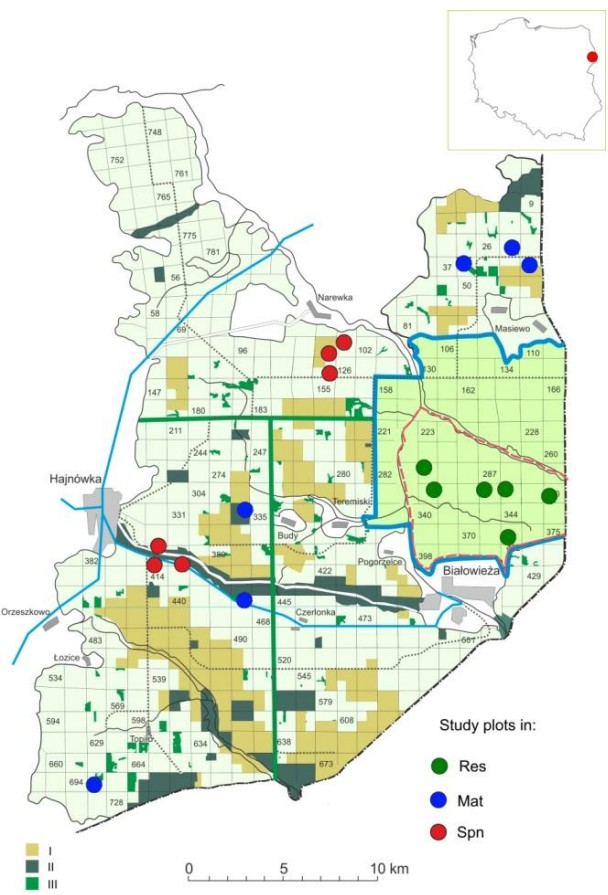

**Figure 1.** Location of study plots in the territory of Białowieża Forest. I—the "Natural Forests of the Białowieża Primeval Forest" reserve called into being in 2003, II—reserves called into being before 2002, III—natural forests outside reserves. Res—old-growth stands in Strict Reserve, Mat—mature stands in managed forests, Spn—spontaneously developed stands.

The mature forest study plots in the managed part of Białowieża Forest (Mat) were located in the following management units: 27Df, 38Aa, 41Ac, 334Ab, 442Bd, and 695Ca. The age of the oldest trees in the plots varied from 160 to 240 years (approximately 200 years on average). The tree stands were predominately composed of oak (20%–60%), hornbeam (0–40%), and spruce (0–30%), with admixtures of Scots pine, aspen, linden, and birch (Figure S2). The total coverage of the first tree layer varied from 30% to 100% (70% on average). In two stands, a second hornbeam layer was present with coverage of 90%–100%. The tree understory was dominated by hornbeam and hazelnut with an admixture of

spruce and oak. It covered approximately 20%–80% (usually 60%) of the area. The ground flora was typical of phytocenoses on fertile soil, similar to plots in the nature reserve. These stands were subjected to the effects of minor silvicultural practices, i.e., sanitary cuts, but no harvesting cuts. In 1998, all stands in the entire Polish part of Białowieża Forest older than 100 years were placed under protection.

The study plots of stands that had spontaneously developed on large-scale clear-cuts (Spn) were located in the following management units: 100Da, 101Ag, 125Ba, 386Bb, 386Ca, and 387Ca. The spontaneous stands in most cases consisted of one tree layer with a dominance of silver birch (40%–60%), spruce (0–40%), or oak (0–10%) and other species, such as hornbeam, aspen, maple, pine, and linden (Figure S3). The total coverage of the first layer varied from 70% to 90% (80% on average). The dominant trees were approximately 70 years old. In two stands, a second layer of trees composed of hornbeam, linden, and spruce was present with coverage of 30%–60%. The understory layer consisted of the youngest hornbeams with an admixture of spruce or common hazel (*Corylus avellana*) with 20%–70% coverage (usually 20%). The spontaneous stands entered the transition phase with the presence of light-demanding pioneer trees in the first story and a well-established layer of shade-tolerant species, mainly hornbeam, in the understory. The ground flora was similar to that observed in the nature reserve, with a slightly lower coverage of *Anemone nemorosa* and higher coverage of *Oxalis acetosella*. These spontaneous stands underwent normal silvicultural practices, such as thinning and sanitary cuts, when needed.

### 2.2. Data Collection and Analysis

The samples for soil fauna extraction were collected from each study plot in May, June, and October 2000. At each of these times, five soil samples per plot were collected with a split soil cylinder (diameter of 5 cm) to 15 cm depth, placed in a plastic bag, and transported to the laboratory. The samples were extracted using a simplified Tullgren apparatus for at least 10 days, and the obtained soil fauna were stored in alcohol. All Collembola specimens were sorted and identified; relatively small specimens were examined on microscopic slides under 40–100× magnification (Axiolab Zeiss, Warsaw, Poland, City, Country phase contrast), while larger specimens were observed under binocular magnification at 10–50× (Olympus SZX9, Warsaw, Poland, City, Country).

Springtails were identified to the species level according to Babenko et al. [42], Fjellberg [43,44], Pomorski [45], Bretfeld [46], Potapov [47], Thibaud et al. [48], and Dunger and Schlitt [49]. According to Gisin [28] and Potapov et al. [29], all species may be classified into one of four different life forms: atmobiotic, epedaphic, hemiedaphic, and euedaphic species. These life forms differ in their fundamental ecological properties, including their vertical distribution, dispersal ability, reproduction, and metabolic activity [50], and can thus be considered as different functional groups [29]. We based our life forms classification on that of Potapov at al. [29] (see Appendix A). Reference specimens were kept at the Department of Forest Protection and Ecology at Warsaw University of Life Science.

The collembolan assemblages in the different forest stand types, i.e., spontaneous (Spn), mature (Mat), and reserve (Res), were compared according to their species diversity, species composition, and life-form structure. First, the completeness of the collembolan assemblage list was checked with an estimator of sample coverage [51]. Species diversity was estimated using Hill numbers of order 0 ($^0$D species richness), 1 ($^1$D exponential of Shannon's entropy index) and 2 ($^2$D inverse Simpson's index) [52]. We present the calculated values on the charts of the individual-based species accumulation curves in each of the forest types and associated 95% unconditioned confidence intervals. For the above analysis, we used the iNEXT package [53] in R 3.6.1 [54].

Differences in the composition of the collembolan assemblages by stand type were visualized using non-metric multi-dimensional scaling (NMDS) with a dissimilarity matrix calculated with the Bray–Curtis index according to the square root transformed number of individuals in each plot. The square root transformation was performed to reduce the effects of the dominant species. Species that were represented by fewer than five individuals in total and observed in fewer than three plots were excluded from the NMDS calculation. Juvenile forms of Collembola were included in the analysis,

since they composed a substantial part of the assemblages. To perform the ordination, the metaMDS function in the vegan package [55] in R was used [54]. Heterogeneity of variances was tested with the adonis and betadisp functions in the vegan package [55]. To confirm the results of the NMDS analysis, we conducted one-way permutational multivariate analysis of variance (PERMANOVA) with the same set of dissimilarity matrices [56]. A pairwise test was performed to determine which assemblages significantly differ. We chose to perform PERMANOVA, since this procedure is generally robust to a moderate heterogeneity of variance in balanced study designs [57]. A species indicator analysis (IndVal) was used to identify species that were significantly associated with specific habitat types [58]. We applied indices modified by De Cáceres et al. [59] using the multipatt () function in the indicspecies 1.7.5 package.

The proportions of life forms in each forest type were analyzed on the basis of a contingency table containing the number of individuals belonging to a particular life form. A chi-square test of independence was used to evaluate whether the number of individuals of different life forms varied among the different forest types. Since significant results were obtained, the standardized residuals were examined as a post hoc test to determine which habitat type and life form drove the differences [60].

## 3. Results

We collected a total of 7711 Collembola specimens, accounting for 91 taxa. Some juvenile forms were identified to the generic level or, in a few cases, to a higher taxonomic level (Appendix A). The sample coverage for all habitats showed a value of c, suggesting that we sampled a substantial proportion of the species present in the area. The highest number of taxa was observed in the old-growth stands in the nature reserve (63), a lower number was found in the plots in mature stands (59), and the lowest number was observed in the spontaneous stands (52). The species accumulation curves of the zero-order Hill numbers displayed high overlap (Figure 2). However, the highest diversity in the extrapolated parts of the accumulation curves was observed for Mat and Res, and the lowest was observed for Spn. This pattern was also confirmed for the first- and second-order Hill numbers (Shannon's and Simpson's indices, respectively); the differences were statistically significant, with the highest diversity observed for the Mat stands and the lowest observed for the Spn stands.

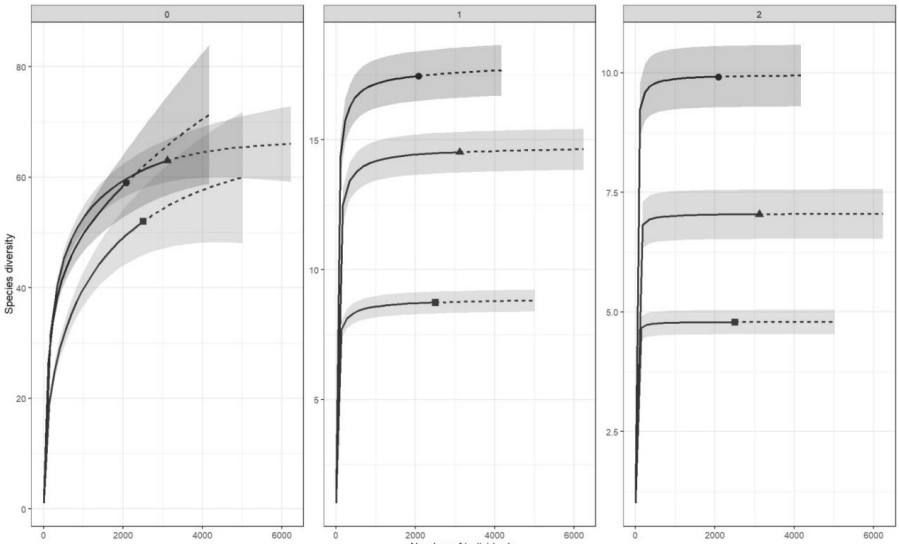

**Figure 2.** Accumulation curves of the Hill numbers of the collembolan assemblages in different forest types. ■ Spn—spontaneously developed stands, ● Mat—mature stands in managed forests, ▲ Res—old-growth stands in Strict Reserve. 0—$^0$D species richness, 1—$^1$D exponential Shannon's entropy index, 2—$^2$D inverse Simpson's index.

The results of the NMDS ordination revealed that the collembolan assemblages in the Spn stands differed from those in the Mat and Res stands in terms of species composition (Figure 3). The assemblages in the Spn stands are mostly positioned on the left side of the diagram, while those in the Mat and Res stands are far to the right, with high overlap between them. The assemblages in the Spn stands showed the lowest average distance to the median, which was equal 0.23, while for Mat and Res, the average distance reached 0.29 and 0.28, respectively ($p < 0.05$). The differences in the assemblages were statistically significant (PERMANOVA $f = 4.32$ $p = 0.001$). The pairwise tests confirmed the significance of the differences between the Spn and Mat assemblages and the Spn and Res assemblages ($F = 6.61$ $p = 0.006$; $F = 8.29$ $p = 0.012$, respectively). The difference between the Mat and Res assemblages was not significant.

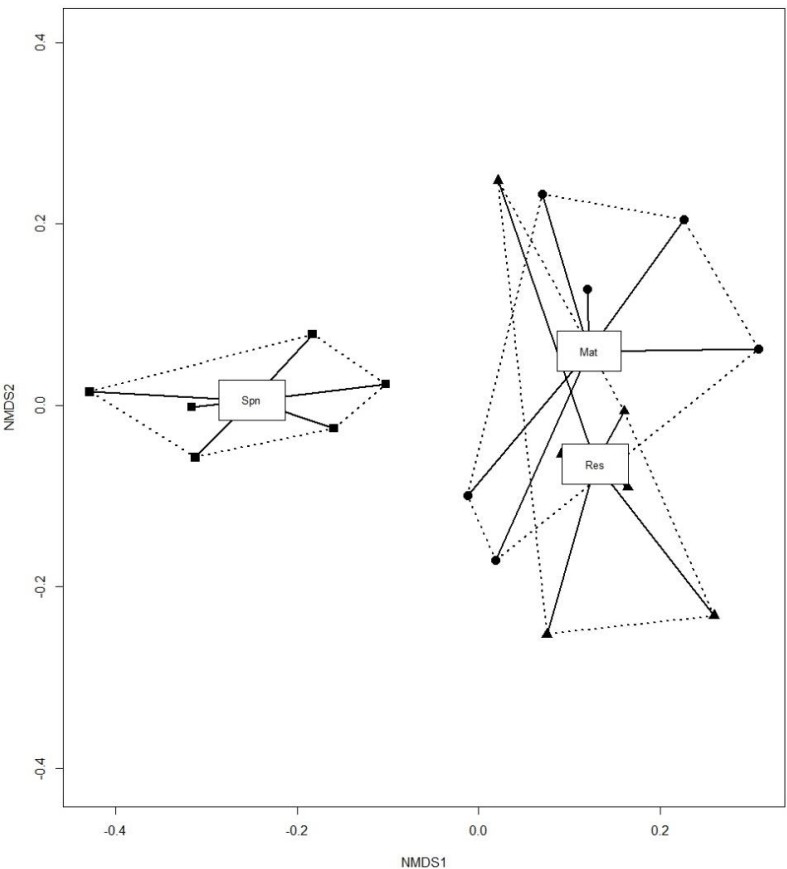

**Figure 3.** Non-metric multi-dimensional scaling (NMDS) ordination plot of the dissimilarities in the collembolan assemblages in different forest types. ■ Spn—spontaneously developed stands, ● Mat—mature stands in managed forests, ▲ Res—old-growth stands in Strict Reserve. Stress 0.18.

The IndVal indicator index values were significant for only seven species (Table 1). Three of them (*Microphorura absoloni*, *Mesaphorura yosii*, and *Arrhopalites spinosus*) were associated with only one forest type, i.e., Res. Another three species (*Pogonognatellus flavescens*, *Lepidocyrtus lignorum,* and *Protaphorura subarmata*) were associated with two forest types, i.e., Res and Mat. Only one species (*Protaphorura pannonica*) had significant IndVal indicator index values for both Spn and Mat stands. The general structure of life-form occurrence among all the studied plots together revealed that their expected proportions in assemblages were as follows: atmobiotic 2%, epedaphic 17%, hemiedaphic 45%, and euedaphic 36%. The life-form structure of the collembolan assemblages differed significantly from the expected structure in all studied stand types (Chisqu = 475.8 *df* = 6 *p* < 0.0001). The percentage of hemiedaphic Collembola was the highest in the Spn stands (60%) and distinctly decreased in the Mat stands (41%) and the Res stands (36%). A contrasting trend was observed in the case of the euedaphic forms, which constitute 28% of the assemblages in Spn, 32% in Mat, and 45 in Res (Table 2).

**Table 1.** Indicator species analysis of the collembolan assemblages in the different forest habitat types. Spn—spontaneously developed stands, Mat—mature stands in managed forests, Res—old-growth stands in Strict Reserve. Only significant results are displayed.

| Indicator Species | Forest Type | Specificity | Fidelity | IndVal Test Statistics | Significance |
|---|---|---|---|---|---|
| *Micraphorura absoloni* | Res | 0.74 | 1 | 0.861 | 0.014 |
| *Mesaphorura yosii* | Res | 0.85 | 0.83 | 0.843 | 0.014 |
| *Arrhopalites spinosus* | Res | 0.93 | 0.67 | 0.787 | 0.011 |
| *Pogonognatellus flavescens* | Mat + Res | 0.97 | 0.92 | 0.945 | 0.023 |
| *Lepidocyrtus lignorum* | Mat + Res | 1 | 0.83 | 0.913 | 0.004 |
| *Protaphorura subarmata* | Mat + Res | 1 | 0.67 | 0.816 | 0.036 |
| *Protaphorura pannonica* | Spn + Res | 0.8651 | 1 | 0.93 | 0.019 |

**Table 2.** Life-form structure of collembolan assemblages in spontaneously developed stands (Spn), mature stands in managed forests (Mat), and old-growth stands in Strict Reserve (Res) with results of pairwise comparisons (post hoc test after $\chi2$ test). Residual values in brackets. Percentages are presented for readability. Total N—number of specimens.

| Life Form | Spn | Mat | Res | Total *N* |
|---|---|---|---|---|
| Atmobiotic | 1% (−2.07)NS | 4% (7.66)*** | 1% (−4.96)*** | 143 |
| Epedaphic | 11% (−9.83)*** | 24% (−9.62)*** | 17% (0.67)NS | 1299 |
| Hemiedaphic | 60% (17.73)*** | 41% (−4.76)*** | 36% (−12.61)*** | 3476 |
| Euedaphic | 28% (−10.12)*** | 32% (−4.72)*** | 45% (13.93)*** | 2793 |
| Total *N* | 2507 | 2086 | 3118 | 7711 |

***$p < 0.0001$; **$p < 0.001$; *$p < 0.05$; NS nonsignificant.

## 4. Discussion

In our study, the recovery of collembolan assemblages during natural forest regeneration after large-scale clear-cutting was assessed for the first time from the perspective of many decades following disturbance. Our study demonstrated that the taxonomic and functional structure of the collembolan assemblages in tree stands that had developed spontaneously in large-scale harvesting plots distinctly differed from the assemblages in old-growth deciduous forests in this region. Seven decades after clear-cutting, the species diversity of the assemblages in the naturally regenerated tree stands was significantly lower, and their structure was altered in comparison to those in the reference forests. There are two explanations for the revealed divergence in collembolan assemblages: (1) large-scale tree harvesting had a strong negative effect on the springtails inhabiting the forest soil, lasting for many decades, and (2) collembolans strongly respond to vegetation type. Thus, the assemblages in naturally regenerated forest stands, which are composed largely of pioneer tree species, differed from those in old-growth forests, in which late-successional tree species dominate.

The disturbance caused by tree harvesting in an area of 10 or more hectares has many potential abiotic and biotic effects on forest ecosystems. The disappearance of the tree canopy results in an increase in extreme temperatures and more frequent drying and wetting at the top of the soil [9]. Logging operations and biomass removal cause forest floor damage, leading to the loss of microhabitats, a modified soil microclimate, and nutrient depletion [14]. According to Covington [61], the organic matter and nutrient content in hardwood stands during the first 15 years after clear-cutting declined by over 50%. In our study of large areas of clearing, unfavorable conditions for litter and soil microarthropods could be reinforced by wind and water erosion.

The detrimental effects of clear-cutting on the diversity and abundance of soil Collembola assemblages have been reported by many authors [9,10,12–15,62–65]. Moreover, it was proven that changes in soil microarthropod communities depend on the management intensity, i.e., biomass removal treatment, forest floor disturbance, or disruption of the soil profile during site preparation [13–15,66].

The strong response of collembolan assemblages to these disturbances results from the fact that the majority of species are very sensitive to soil moisture and temperature changes, and their development and reproduction strongly depend on these factors [20]. It is worth mentioning that forest floor collembolans are very vulnerable even to small-scale litter depletion [67].

In the evaluation of the ecological consequences of clear-cutting, the key question is how long the effects of these types of disturbances for forest microarthropods persist. In view of the information provided by the literature, no generalization can be made concerning the time necessary for the recovery of forest collembolan assemblages after stand harvesting. First, the majority of studies have been performed in managed coniferous forest subjected to a clear-cutting silvicultural system for many forest rotation cycles. Management practices cause the simplification and unification of stand structure, and could modify the course of the regenerative succession of soil biota. Second, the harvesting of managed forest stands is usually confined to a few hectares, and in recent decades, such areas are even more restricted in size. The results of an experiment conducted in plots that were 1 hectare in size reported the lack of a response of soil springtails to traditional clear-felling [68]. Third, in most cases, the soil fauna were studied shortly after the felling activity, when the disruption of forest ecosystems causes strong instability in the number and abundance of many forest floor taxa. Huhta et al. [69,70] and Huhta [71] reported initial increases in mite and springtail densities after clear-felling, followed by significant declines. Thus, the results, even in the case of coniferous forests, are strongly dependent on the time lag since harvesting and size of clearings. Additionally, in managed forests, different methods of site preparation and planting can blur the response of soil fauna to harvesting and delay their recovery [35,72].

The persistent modification of collembolan community structure as a result of clear-cutting has been reported mainly in relation to coniferous forests. Addison [17] documented a lower density and species richness and altered community structure of the collembolan community 10 years after harvesting in the spruce forests of British Columbia. According to Bengtsson et al. [18], the changes in the soil food web after whole-tree harvesting may be very long term. In a Swedish Scots pine forest at a poor site, a decrease in Collembola density of 35% lasted for 17 years. However, in the same study, no significant effect of clear-cutting was found in more productive soil planted with spruce forests. The lack of a response of the soil fauna in the spruce plantation was likely due to the accumulation of organic matter and nutrients on the forest floor, which can partly buffer microclimate modification in clear-cut areas and foster the recovery of soil microarthropods [68]. The incomplete recovery of Collembola communities in pine forests in northeastern Ontario 20 years after clear-cutting was reported by Rousseau et al. [19] in a study that involved complementary taxonomic and trait-based approaches, which is currently the longest-term experiment on mesofaunal communities performed in permanent plots in coniferous stands. However, studies of successional changes in Collembola communities during spruce forest rotation using chronosequences of stands indirectly proved that mesofaunal communities need a very long time to recover after harvesting. The clear impoverishment of communities and changes in their structure identified in 45-year-old secondary tree stands indicate that the effects of clear-cutting are persistent [36].

The results of studies conducted in coniferous forests do not correspond with those in deciduous forests, because these forest types differ considerably with respect to litter composition and structure, humus form, mesofauna communities, and forest floor processes [73–77]. Moreover, the impact of clear-cutting on the soil biota of deciduous forests and the postharvest recovery of these organisms have seldom been studied, because other management methods are applied in these forests. However, the destructive effects of clear-cutting on soil mesofauna, including springtails, has been reported. In the first year following clear-cutting in Appalachian hardwood forest, the mean density of litter microarthropods was reduced by >50%, and because the various taxa responded differently, the relative abundance of major groups was changed [62]. The results of a study conducted over eight years of forest regrowth indicated only a partial recovery of the faunal density and that the Collembola abundance was 24% lower than that in the reference uncut forest [63]. Impoverishment of the Collembola assemblage

after clear-cutting was also reported in deciduous forests in Japan, but in this case, the species richness and abundance of collembolans recovered four years after harvesting [78]. This very fast recovery of the soil fauna was attributed to the prompt redevelopment of the deciduous plant communities in the studied plots, where some shrub and small tree species, such as *Rubus* spp. and *Salix* spp., immediately invaded the clear-cut areas, and some tree species (e.g., *Quercus serrata*) regenerated from stumps by sprouting. The rapid appearance of young secondary forest, which creates favorable conditions for the growth and reproduction of soil microarthropods, can explain the rapid recovery of the soil biota.

In the case of our study, because of the large area of the harvested plots, abiotic factors could have hindered the progress of vegetation colonization and establishment. As a result, the forest floor microarthropods, which were exposed to unfavorable conditions in open areas for a relatively long period of time, were decimated, and the structure of their assemblages changed over a much longer time period. Studies of Collembola in forest stands in the High Tatra Mountains after a catastrophic windthrow partly prove this hypothesis [79]. The negative effects of large-scale clear-cutting and the extraction of fallen wood on mesofauna decreased over time, but remained apparent after five years in the case of edaphic Collembola, because several species sensitive to deforestation and clear-cutting are absent in these communities.

The second explanation for the distinct differences in the collembolan assemblages in the naturally regenerated forest stands and the old-growth reference forests revealed by our study is that collembolans strongly respond to vegetation type. The tree stand composition in all study plots was believed to be similar before clear-cutting, because all of the stands developed as a result of natural processes on similar soils. The replacement of these forest stands dominated by late-successional tree species by spontaneously colonizing pioneer tree species after harvesting reshaped the structure of the forest ecosystem and changed the quantity and quality of the litter entering the forest floor, rhizosphere inputs, and ground vegetation. However, whether all these changes modified the soil fauna is a matter of debate [80]. All possible reactions—positive, negative or no effect—were found when the effects of tree stand replacement on the diversity and abundance of microarthropods were studied. As a consequence of this variety in response, no overall significant differences in the species richness of Collembola among the tree stand types were found, which was most likely due to site-specific influences and high variability. On the other hand, some studies have shown that forest-stand differences considerably affected the composition of collembolan assemblages, but only along a gradient of deciduous to coniferous stands [76,81]. This is not the case in our study plots, because all of them were located at one point along this gradient: mixed deciduous forests on Cambisols. However, the relatively young forest stands that spontaneously developed in the harvested plots differed structurally from the old-growth forest stands with respect to the organic matter that had accumulated on the forest floor, particularly in terms of the coarse and fine woody debris. Therefore, the persistent difference in the structure of the Collembola assemblages between these forests was possibly partly due to the reduced presence of deadwood in the younger forests, since decaying wood provides essential microhabitats with heterogeneous spatial structure, a stable microclimate, and various food resources, such as fungi. Many collembolan species are highly dependent on decaying wood of various stages [82], and the presence of dead wood of various sizes has potential long-term benefits for mesofaunal communities [19]. Unfortunately, this aspect was omitted in our studies, but as the volume of deadwood in forest ecosystems increases over their succession [83], we can assume that this structural element of tree stands is indirectly reflected by the age of the forest.

Therefore, we suggest that the impoverishment of Collembola assemblages in spontaneously regenerated tree stands in plots harvested seven decades ago is primarily a legacy of the clear-cutting of the large area of forest that had never been subjected to such disturbance. Clear differences in the taxonomic structure of the collembolan assemblages between the stands that had spontaneously developed in the harvested plots and all the studied mature and old-growth forests were revealed using NMDS. This result corresponds with the outcome of the IndVal analysis, because species such as *Microphorura absoloni*, *Mesaphorura yosii*, *Arrhopalites spinosus*, *Pogonognatellus flavescens*, *Lepidocyrtus*

*lignorum,* and *Protaphorura subarmata* occurred at the highest frequency in the reference forests. Moreover, among the indicative taxa, the species belonging to the euedaphic life form were dominant, so in the mature forest and old-growth stands, they are a distinctive component of the assemblages. In stands that have developed on large-scale clearings, the share of euedaphic forms of Collembola in assemblages was lower than that in reference forests. Species of this life form display low dispersal ability because of their anatomical features, such as their small body size, short legs, and lack of furcula. They can also be prone to the unfavorable conditions associated with clear-cutting, such as reduced soil moisture and elevated temperature [63], soil compaction [11], and limited food resources [84]. In general, the response of the various life forms of Collembola to clear-cutting is site-dependent because, for example, the abundance of the epedaphic form can increase, decrease, or remain unchanged after harvesting depending on the time that has passed following the treatment, site productivity, and management intensity [9,12–14,18,36]. Usually, these species are the first to disappear shortly after harvesting, especially in the case of intense biomass removal and forest floor disruption [19]. Then, some of them are able quickly recolonize disturbed areas due to their dispersal ability and the presence of functional traits allowing them endure the unfavorable conditions of open space [19,85]. The large share of hemiedaphic forms of collembolans in the assemblages in spontaneously developed stands observed in our study is in accordance with the results of Farská et al. [13]. In the mesofaunal communities associated with intensively managed spruce forests, an increase in hemiedaphic species at the expense of euedaphic and epedaphic species was observed by these authors.

The high prevalence of hemiedaphic forms in the assemblages in disturbed stands led to the different functional structure of these stands when compared to those in the reference old-growth forests, where hemiedaphic forms are balanced with euedaphic forms. This modification of life form structure can have substantial implications for soil processes. In old-growth deciduous forests, the activity of Collembola in the lower litter and mineral soil, which regulate the microbial communities in the rhizosphere and soil organic matter decomposition, are in balance with the Collembola in the upper litter layers, controlling the microbial communities and affecting the dynamics of litter decomposition. The shift in functional group structure in stands that regenerate after clear-cutting can indicate that the activity of collembolan assemblages seven decades after disturbance are mainly concentrated on the decomposition of litter in the upper layers, whereas the processes controlled by these organisms in deeper soil layers are not fully restored. It is likely that harvesting along with biomass removal has had detrimental impacts on the soil physicochemical properties, and has limited the occurrence of euedaphic Collembola for many decades. This treatment significantly increases the bulk density and penetration resistance of the soil, decreases the total soil porosity [86] and, as a result, restricts water and nutrient access to the deeper soil layers and reduces air diffusion [87,88]. Collembolans, similar to many other soil microarthropods, are very sensitive to soil compaction [89,90], but prolonged disruption in assemblage structure has never been reported. This aspect of clear-cutting, because of its broad ecological importance, requires further comprehensive studies.

## 5. Conclusions

We have shown that the collembolan assemblages in 70-year-old deciduous stands that spontaneously developed in large-scale harvesting plots differed substantially from those in deciduous old-growth stands in Białowieża Forest. The species diversity of the assemblages in the disturbed areas was significantly lower, and their taxonomic structure was noticeably different from those in the reference forests. Furthermore, the life-form structure was modified and the hemiedaphic forms of Collembola predominated in the assemblages in the spontaneously developed stands. In the old-growth forests, the shares of euedaphic forms and hemiedaphic forms were more balanced. The revealed modification of life-form proportions suggests that the processes driven by Collembola in the deeper soil layers were altered. The role fulfilled by euedaphic Collembola is the consumption of euedaphic microorganisms, which affects nutrient uptake by roots and regulates the microbial community in the rhizosphere and soil organic matter decomposition. The results showed that the

taxonomic and functional recovery of Collembola assemblages after large-scale disturbance is very slow, and after seven decades of succession, these assemblages still differ significantly from those in the reference old-growth forests. Therefore, the negative impacts of activities such as salvage logging, which is usually conducted after windstorms, or bark beetle attacks on soil mesofauna may persist for many decades.

**Supplementary Materials:** The following are available online at http://www.mdpi.com/1999-4907/10/11/948/s1. Figure S1. Stand in strict reserve of Białowieża National Park (photo T. Mokrzycki); Figure S2. The mature forest in the managed part of Białowieża Forest (photo M. Sławski); Figure S3. The stands spontaneously developed on large-scale clear-cuts (photo T. Mokrzycki).

**Author Contributions:** M.S. (Marek Sławski) and M.S. (Małgorzata Sławska); conceptualization and methodology, M.S. (Marek Sławski) and M.S. (Małgorzata Sławska); fieldworks, M.S. (Marek Sławski) and M.S. (Małgorzata Sławska); Collembola identification, M.S. (Małgorzata Sławska); data analysis, M.S. (Marek Sławski); writing—original draft preparation M.S. (Marek Sławski) and M.S. (Małgorzata Sławska).

**Funding:** The State Committee for Scientific Research (grant nr 5 P06H013 15) funded this research, which was conducted under the guidance and direction of Andrzej Szujecki.

**Acknowledgments:** The authors thank the park director, Czesław Okołow, for permitting research in Białowieża National Park and Andrzej Keczyński for selecting the study sites.

**Conflicts of Interest:** The authors declare no conflict of interest.

## Appendix A

Species list, life forms, and total number of individuals of Collembola in stands that spontaneously developed in a clear-cut area (Spn), mature mixed deciduous forest (Mat), and old-growth forest in Strict Reserve of Białowieża National Park (Res). Life forms: atmobiotic (a), epedaphic (ep), hemiedaphic (h), euedaphic (eu). Life form classification according to Potapv at al. [29].

| Taxa | Life Form | Spn | Mat | Res | Total |
|---|---|---|---|---|---|
| *Ceratophysella engadinensis* (Gisin, 1949) | ep | 1 | 56 | 40 | 97 |
| *C. mosquensis* (Becker, 1905) | ep | - | 1 | - | 1 |
| *Ceratophysella* sp. juv. | ep | 3 | 13 | 2 | 18 |
| *Xenylla brevicauda* (Tullberg, 1869) | h | 7 | - | - | 7 |
| *Xenylla* sp. juv. | h | 25 | - | - | 25 |
| *Willemia anopthalma* (Börner, 1901) | eu | 1 | - | 56 | 57 |
| *W. denisi* (Mills, 1932) sensu (Fjellberg 1985) | eu | - | 8 | 28 | 36 |
| *Willemia* sp. juv. | eu | - | 1 | 2 | 3 |
| *Xenyllodes armatus* (Axelson, 1903) | h | - | 1 | - | 1 |
| *Friesea claviseta* (Axelson, 1900) | ep | - | - | 56 | 56 |
| *Friesea* sp. juv. | ep | - | - | 1 | 1 |
| *Pseudachorutes corticicolus* (Schäffer, 1896) | ep | - | 3 | 10 | 13 |
| *P. dubius* (Krausbauer, 1898) | ep | 1 | - | - | 1 |
| *P. parvulus* (Börner, 1901) | ep | 1 | 1 | 11 | 13 |
| *Pseudachorutes* sp. juv. | ep | 3 | - | 3 | 6 |
| *Micranurida pygmea* (Börner, 1901) | h | 1 | - | 9 | 10 |
| *Anurida granulata* (Agrell, 1943) | h | - | 4 | 3 | 7 |
| *Neanura muscorum* (Templeton, 1835) | h | 23 | 5 | 9 | 37 |
| *Micraphorura absoloni* (Börner, 1901) | eu | 5 | 9 | 40 | 54 |
| *Protaphorura bicampata* (Gisin, 1956) | eu | - | 16 | - | 16 |
| *P. pannonica* (Haybach, 1960) | eu | 55 | 17 | 54 | 126 |
| *P. subarmata* (Gisin, 1957) | eu | - | 67 | 67 | 134 |
| *P. tricampata* (Gisin, 1956) | eu | 7 | 1 | - | 8 |
| *Protaphorura* sp. juv. | eu | 16 | 11 | 58 | 85 |
| *Supraphorura furcifera* (Börner, 1901) | eu | - | 11 | - | 11 |
| *Hymenaphorura polonica* (Pomorski, 1990) | eu | - | - | 25 | 25 |

| Taxa | Life Form | Spn | Mat | Res | Total |
|---|---|---|---|---|---|
| *Mesaphorura critica* (Ellis, 1976) | eu | - | 1 | 10 | 11 |
| *M. hylophila* (Rusek, 1982) | eu | 3 | 9 | 7 | 19 |
| *M. italica* (Rusek, 1971) | eu | - | - | 3 | 3 |
| *M. macrochaeta* (Rusek, 1976) | eu | 65 | 54 | 91 | 210 |
| *M. sylvatica* (Rusek, 1971) | eu | - | - | 1 | 1 |
| *M. tenuisensillata* (Rusek, 1974) | eu | - | 7 | 12 | 19 |
| *M. yosii* (Rusek, 1967) | eu | - | 6 | 35 | 41 |
| *Mesaphorura* sp. juv. | eu | 1 | 1 | 9 | 11 |
| *Karlstejnia norvegica* (Fjellberg, 1974) | eu | 1 | - | 1 | 2 |
| *Stenaphorurella quadrispina* (Börner, 1901) | eu | 1 | - | - | 1 |
| *Anurophorus septentrionalis* (Pallisa, 1966) | h | - | - | 30 | 30 |
| *Anurophorus* sp. juv. | h | 1 | - | 33 | 34 |
| *Folsomia dovrensis* (Fjellberg, 1976) | eu | - | 1 | 1 | 2 |
| *F. fimetarioides* (Axelson, 1903) | eu | - | 73 | - | 73 |
| *F. lawrencei* (Rusek, 1984) | eu | 4 | - | - | 4 |
| *F. manolachei* (Bagnal, 1939) | h | 10 | - | - | 10 |
| *F. stella* (Christansen & Tucker, 1977) | eu | - | 1 | 9 | 10 |
| *F. quadrioculata* (Tullberg, 1871) | h | 228 | 420 | 631 | 1279 |
| *Appendisotoma juliannae* (Traser, 1993) | ep | - | - | 190 | 190 |
| *Proisotoma armeriae* (Fjellberg, 1976) | h | - | 1 | - | 1 |
| *P. minima* (Tullberg, 1871) | h | 4 | 5 | 31 | 40 |
| *Proisotoma* sp. juv. | h | - | - | 1 | 1 |
| *Isotomiella minor* (Schäffer, 1896) | eu | 509 | 339 | 893 | 1741 |
| *Parisotoma notabilis* (Schäffer, 1896) | h | 970 | 310 | 323 | 1603 |
| *Vertagopus* sp. juv. | ep | - | - | 1 | 1 |
| *Isotoma viridis* (Bourlet, 1839) | ep | - | 41 | - | 41 |
| *Isotoma* sp. juv. | ep | 2 | 11 | 2 | 15 |
| *Desoria divergens* (Axelson, 1900) | ep | 8 | - | 12 | 20 |
| *D. tigrina* (Tullberg, 1871) | ep | - | 33 | 13 | 46 |
| *D. violacea* (Tullberg, 1876) | ep | - | 44 | 12 | 56 |
| Isotomidae juv. | ep | - | - | 30 | 30 |
| *Tomocerus vulgaris* (Tullberg, 1871) | ep | - | 1 | 2 | 3 |
| *Pogonognathellus flavescens* (Tullberg, 1871) | ep | 4 | 99 | 47 | 150 |
| Tomoceridae juv. | ep | 2 | 12 | 4 | 18 |
| *Orchesella bifasciata* (Nicolet, 1841) | a | 2 | - | - | 2 |
| *O. flavescens* (Bourlet, 1839) | a | 9 | 55 | 10 | 74 |
| *Orchesella* sp. juv. | a | 3 | 3 | - | 6 |
| *Entomobrya corticalis* (Nicolet, 1841) | a | 2 | - | - | 2 |
| *E. nivalis* (Linnaeus, 1758) | a | 2 | - | - | 2 |
| *Entomobryides myrmecophilus* (Reuter, 1886) | a | 2 | 1 | - | 3 |
| *Willowsia buski* (Lubbock, 1869) | a | 1 | - | - | 1 |
| *Willowsia nigromaculata* (Lubbock, 1876) | a | 1 | 4 | 7 | 12 |
| *Lepidocyrtus lignorum* (Fabricius, 1793) | ep | - | 37 | 47 | 84 |
| *Lepidocyrtus* sp. 1 | ep | 2 | - | - | 2 |
| *Lepidocyrtus violaceus* gr juv. | ep | 1 | - | - | 1 |
| *Lepidocyrtus lignorum* gr juv. | ep | 106 | 3 | 24 | 133 |
| *Pseudosinella alba* (Packard, 1873) | h | 65 | - | - | 65 |
| *Pseudosinella zygophora* (Schille, 1908) | h | 156 | 93 | 49 | 298 |
| *Entomobyidae* juv. | ep | 101 | 54 | 4 | 159 |
| *Neelides minutus* (Folsom, 1901) | eu | - | 1 | - | 1 |
| *Megalothorax minimus* (Willem, 1900) | eu | 40 | 33 | 16 | 89 |
| *Sphaeridia pumilis* (Krausbauer, 1898) | ep | 14 | 38 | 2 | 54 |
| *Sminthurides malmgreni* (Tullberg, 1876) | a | - | 2 | - | 2 |

| Taxa | Life Form | Spn | Mat | Res | Total |
|------|-----------|-----|-----|-----|-------|
| *Arrhopalites caecus* (Tullberg, 1871) | h | - | 2 | - | 2 |
| *A. spinosus* (Rusek, 1967) | h | - | 1 | 13 | 14 |
| *Arrhopalites* sp. juv. | h | 3 | 6 | 3 | 12 |
| *Sminthurinus elegans* (Fitch, 1863) | ep | 1 | 2 | - | 3 |
| *Sminthurinus* sp. juv. | ep | 13 | 19 | 9 | 41 |
| *Ptenothrix* sp. juv. | a | - | - | 7 | 7 |
| *Dicyrtoma* sp. juv. | a | - | 1 | 2 | 3 |
| *Lipotrix lubbocki* (Tullberg, 1872) | ep | 8 | 14 | 4 | 26 |
| *Allacma fusca* (Linnaeus, 1758) | a | 3 | - | - | 3 |
| *Caprainea marginata* (Schött, 1893) | ep | - | 10 | 10 | 20 |
| *Sminthuridae* juv. | a | - | - | 2 | 2 |
| *Symphypleona* juv. | a | 10 | 13 | 1 | 24 |
| Total number of individuals | | 2507 | 2086 | 3118 | 7711 |
| Total number of taxa | | 52 | 59 | 63 | 91 |

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
