# Peer review of "Seven Decades of Spontaneous Forest Regeneration after Large-Scale Clear-Cutting in Białowieża Forest do not Ensure the Complete Recovery of Collembolan Assemblages"

_forests, doi:10.3390/f10110948_

Round 1

Reviewer 1 Report

This is an original and a very important contribution on long-term succession of Collembola assemblages in secondary spontaneous vegetation after extensive historical clear-cuts in the Bialowieza Forest. Both authors are very well skilled in such kind of studies, the second one is guarantee of the proper taxonomical identification of Collembola as the model group. The study had a well established design with six replicate forest plots for each of the three management treatments – (1) old-growth stands in the reserve, (2) managed mature forest study plots, and (3) forests spontaneously developed on large-scale clear-cuts.   

The number of soil samples collected and the material analysed is fully sufficient for such kind of studies. The authors analysed communitry data using modern statistics and multivariate methods and came to very important results, especially to documentation that even during long-term spontaneous succession of the vegetation, lasting several decades, Collembola assemblages do not fully recover compared to old, intact mixed forests in the Bialowieza National Park.

I consider this manuscript as very suitable for publishing in the journal Forests after minor corrections, marked directly in the manuscript pdf file, and my recommendations provided below.

The only serious corrections are addressed to reconsideration of several species regarding their life forms in Appendix and subsequent recalculation of their share in communities and comparison between different types of forest stands. Moreover, some collected specimens were juveniles, which were therefore sorted to higher taxa (genera, families). Since most Collembola  develop in the soil during juvenile instars, they should be considered edaphic forms (hemi- or euedaphic).  Regarding this, specify please in the section "2.2. Data collection and analysis" the references which you followed for specification of life forms in individual species. I guess the principal sources were the same as for species identification. I also recommend to follow the source:

Rusek J., 2007: A new classification of Collembola and Protura life forms. In: Tajovský K., Schlaghamerský J., Pižl V. (eds.) Contributions to Soil Zoology in Central Europe II. Biological Centre AS CR, Institute of Soil Biology, České Budějovice, 109–115.

Here I provide some points that may substantially improve the paper for the readers:

1) I recommend to show locations of the research plots on a situation map.

2) The forests stands are well described, nevertheless, documentary figures from the research plots would provide a reader better idea of them.

3) The paper is linguistically well written, nevertheless, I recommend to check it by a native English speaker.

Author Response

We appreciate and thank for the constructive criticism and detailed remarks.

Point 1: The only serious corrections are addressed to reconsideration of several species regarding their life forms in Appendix and subsequent recalculation of their share in communities and comparison between different types of forest stands. Moreover, some collected specimens were juveniles, which were therefore sorted to higher taxa (genera, families). Since most Collembola  develop in the soil during juvenile instars, they should be considered edaphic forms (hemi- or euedaphic).  Regarding this, specify please in the section "2.2. Data collection and analysis" the references which you followed for specification of life forms in individual species. I guess the principal sources were the same as for species identification. I also recommend to follow the source: Rusek J., 2007: A new classification of Collembola and Protura life forms. In: Tajovský K., Schlaghamerský J., Pižl V. (eds.) Contributions to Soil Zoology in Central Europe II. Biological Centre AS CR, Institute of Soil Biology, České Budějovice, 109–115.

Response 1:

The life forms was defined as trait complex composed of number of ommatidia, body size and intensity of coloration (Gisin, 1943). Additional criteria such as leg length, presence or absence of furca and its length was used by Stebaeva (1970) to improve the life form concept. The classification of collembolan species into different forms was and still is a matter of debate (Petersen 2002, Chahartaghi et al. 2006, Rusek 2007, Vandewalle et al. 2010, da Silva et al. 2016). Recent studies using stable isotope technic have considerably increased knowledge on collembolan life forms and their trophic niches (Potapov et al. 2017). Moreover, in these studies over eighty species of temperate forest ecosystems was tested and assigned into life forms following Stebaeva (1970) and Rusek (2007); see title of Table 1 in Potapov et al. (2017). In other words classification that we used after Potapov at al. (2017)  is updated and revised version of the Rusek (2007) and Stebaeva (1970) proposals.

In our paper in the section "2.2. Data collection and analysis" we wrote: According to Gisin [28] and Potapov et al. [29], all species may be classified into one of four different life forms: atmobiotic, epedaphic, hemiedaphic and euedaphic species (see Appendix) and in our opinion it is a clear indication of references which we use for specification of individual taxon into life forms. To be more clear we added precise information about the literature source of the classification we used in the section "2.2. Data collection and analysis"  and in the title of the table in the Appendix as well.

We do not agree with reviewer that juvenile instars of Collembola should be considered as edaphic forms (hemi- or euedaphic). According to recent literature (Pollierer and Scheu 2017) all juvenile specimens should be classified as adults ones belonging to the same genus. For example: Ceratophysella sp./juv. was assigned by these authors to epedaphic life form like as C.armata, C.bengtssoni, C.denticulata, C.gibbosa and C.luteospina. Similarly Xenylla juv.,  Isotoma juv., Lepidocyrtus juv. and Smithurides juv. were also assigned to epedaphic. Also juvenile instars determined to higher taxa such as Sminthuridae was assigned to epedaphic life form as the majority of representatives of these taxa and analogically Entomobyidae juv. to atmobiotic. For more details see Table A1  in Pollierer & Scheu (2017).

References

Chahartaghi, M., Scheu, S., Ruess, L. 2006. Sex ratio and mode of reproduction in Collembola of an oak-beech forest. Pedobiologia, 50(4), 331-340. Gisin, H. (1943). Okologie und Levensgemenischaften der Collembolen im schweizerischen Exkursionsgebiet Basels. Revue suisse de Zoologie, 50, 131-224. Potapov, A. A., Semenina, E. E., Korotkevich, A. Y., Kuznetsova, N. A., & Tiunov, A. V. 2016. Connecting taxonomy and ecology: Trophic niches of collembolans as related to taxonomic identity and life forms. Soil Biology and Biochemistry, 101, 20-31. Petersen, H. 2002. General aspects of collembolan ecology at the turn of the millennium: Proceedings of the Xth international Colloquium on Apterygota, České Budějovice 2000: Apterygota at the Beginning of the Third Millennium. Pedobiologia, 46(3-4), 246-260. Pollierer, M. M., Scheu, S. 2017. Driving factors and temporal fluctuation of Collembola communities and reproductive mode across forest types and regions. Ecology and evolution, 7(12), 4390-4403. Rusek J., 2007. A new classification of Collembola and Protura life forms. In: Contributions to Soil Zoology in Central Europe II. Biological Centre AS CR, Institute of Soil Biology, České Budějovice, 109–115. da Silva, P. M., Carvalho, F., Dirilgen, T., Stone, D., Creamer, R., Bolger, T., Sousa, J. P. 2016. Traits of collembolan life-form indicate land use types and soil properties across an European transect. Applied soil ecology, 97, 69-77. Stebaeva S.K.1970. Life forms of springtails (Collembola). Zool. Zhurnal 49: 1473-1455 (in Russian). Vandewalle, M., De Bello, F., Berg, M. P., Bolger, T., Doledec, S., Dubs, F., Da Silva, P. M. 2010. Functional traits as indicators of biodiversity response to land use changes across ecosystems and organisms. Biodiversity and Conservation, 19(10), 2921-2947.

Point 2: I recommend to show locations of the research plots on a situation map.

Response 2: Done.

Point 3: The forests stands are well described, nevertheless, documentary figures from the research plots would provide a reader better idea of them.

Response 3: Done in Supplement.

Point 4: The paper is linguistically well written, nevertheless, I recommend to check it by a native English speaker.

Response 3: The certificate of AJE was enclosed.

Line 48-49   Białowieża Forest, inscribed by UNESCO on the World Heritage List, includes a complex of lowland forests that are characteristic of the Central European mixed forests terrestrial ecoregion.

Provide here the adequate literature source in brackets at the end of sentence. 

Response: The uniqueness of Białowieża Forest nature, its outstanding natural values for biodiversity conservation, representative exemplification of on-going ecological and biological processes of great importance in ecosystem evolution and progress, as well as scientific values unparalleled have been recognized on the international forum.

Line 77-80   In temperate and boreal forests, springtails (Collembola, Hexapoda) represent some of the most numerous soil microarthropods involved in organic matter decomposition, nutrient cycling and soil microstructure improvement [20]. A large number of collembolan species occurring in litter and soil contribute essentially to the biodiversity of forest ecosystems and, as a result, enrich the diversity and functionality of belowground food webs [21–24].

ReErase this paragraph since it is about the well-known and often repeated facts about Collembola, and moreover beyond the scope of this contribution.

Response: In our opinion these facts are obvious for soil researchers, but we are not convinced that these knowledge are common among forester or ecologist.  

Line 142-143  The study plots in the reserve (Res) were located in the following management units: 284Ba,

314Bd, 317Ab, 318Df, 320Cc, and 373Db.

Line 158, 159  The mature forest study plots in the managed part of Białowieża Forest (Mat) were located in

the following management units: 27Df, 38Aa, 41Ac, 334Ab, 442Bd, and 695Ca.

Line 170, 171   were  located in the following management units: 100Da, 101Ag, 125Ba, 386Bb, 386Ca, and 387Ca

This is not important for the reader. Instead, you should provide a map of the area with location of the study plots.

Response: We think that detailed information about location of study plots is very important, especially in studies on succession in which subsequent studies are highly desirable.

Line 228, 229

Since significant results were obtained, the standardized residuals were examined as a post hoc test to determine which habitat type and life form drove the differences [60].

I recommend to recategorize life form specification in several taxa - see my comments in Appendix.

Response: see explanations to Point 1

Line 235-237 

I recommend to provide abbreviations of three types of stands in this sentence that would be clearer for the reader when reading the following sentence. 

Response: The full names of  three types of stands were used only at the beginning of the section 3.Results to make results clearer for the reader. Later only the abbreviations were applied.

Line 267-274  The proportion of individual life forms and their differences between stands should be recalculated. 

Response: see explanations to Point 1

Line 302  Specify here what do you mean with unfavourable conditions.

Response: unfavorable conditions were described above in lines 297-299: The disappearance of the tree canopy results in an increase in extreme temperatures and more frequent drying and wetting at the top of the soil [9]. Logging operations and biomass removal cause forest floor damage, leading to the loss of  microhabitats, a modified soil microclimate and nutrient depletion [14].

Line 370  Provide here some literature sources.

Response: In line 374 we provide the reference [79]

Remarks in Appendix.

Response: see explanations to Point 1

Reviewer 2 Report

The authors laid out a strong and compelling case. This paper was well articulated and developed. Nice job!

Author Response

We appreciate and express gratitude for your recommendation.